# Barriers to and Facilitators of Physical Activity among Korean Female Adults with Knee Osteoarthritis and Comorbidity: A Qualitative Study

**DOI:** 10.3390/healthcare8030226

**Published:** 2020-07-23

**Authors:** Eunyoung Park, Hyung-Ran Park, Eui-Sung Choi

**Affiliations:** 1College of Nursing, Chungnam National University, Jung-gu, Munhwa-ro 266, Daejeon 35015, Korea; eypark@cnu.ac.kr; 2Department of Nursing Science, College of Medicine, Chungbuk National University, Chungdae-ro 1, Seowon-Gu, Cheongju, Chungbuk 28644, Korea; 3Department of Orthopedic Surgery, College of Medicine, Chungbuk National University, Chungdae-ro 1, Seowon-Gu, Cheongju, Chungbuk 28644, Korea; oseschoi@chungbuk.ac.kr

**Keywords:** osteoarthritis, comorbidity, physical activity, qualitative research

## Abstract

When knee osteoarthritis is combined with comorbidity, it is associated with limited physical activity. This study aimed to identify barriers to and facilitators of physical activity among Korean female adults with knee osteoarthritis and comorbidity, such as hypertension, diabetes, and dyslipidemia. A qualitative content analysis study was conducted. Ten female knee osteoarthritis participants with comorbidity were recruited at an orthopedic outpatient center in South Korea. Data were collected using in-depth interviews and were analyzed using a conventional content analysis method. Ten participants with a mean age of 70.7 years participated in this study. Four categories of barriers and three of facilitators were identified. Barriers to physical activity were physical hardships, lack of motivation, environmental restrictions, and lack of knowledge. Categories of facilitators were pain management, self-control in physical activity, and understanding the importance of physical activity. Participants did not express any social or environmental facilitators of physical exercise. Healthcare professionals should include social support and environmental facilities to achieve medical and institutional compliance. Understanding female adults with knee osteoarthritis and comorbidity would support provision of appropriately tailored interventions that account for the characteristics of the comorbidity.

## 1. Introduction

Osteoarthritis (OA) is a degenerative joint disease that is increasingly prevalent because populations are aging [1,2]. Nearly 40% of Korean adults over 50 years of age have knee OA [3,4] and OA is the second most chronic disease in the Korean health care utilization rate, which increases the economic burden of the Korean health care system [5]. The majority (70–80%) of Korean adults with knee OA are women [3,4]. Loss of estrogen in perimenopausal time effects the knee joint and increases knee OA in women [6].

Currently, patients with knee OA have a relatively high likelihood of developing comorbidity and, consequently, many knee OA patients are diagnosed with two or more diseases [3,7]. Most Korean adults with knee OA have other chronic diseases; conditions of high frequency are hypertension (46.8–54.5%), dyslipidemia (17.8–19.7%), and diabetes (15.1–17.4%) [3]. Prevalence of other chronic conditions is under 7% [3]. Previous studies on comorbidity in knee OA patients have also found evidence that hypertension, diabetes, and dyslipidemia are the key risk diseases for knee OA [4,8].

OA is characterized by pain and inflammation of the articular cartilages leading to limitations on ordinary physical activities [8,9]. Approximately 83.2% of Korean female adults with knee OA experience pain, and knee OA adults with pain experience 2.83 times more physical activity limitations than adults without knee OA [3]. For knee OA, physical activity is recommended in international clinical guidelines to relieve pain level and to enhance function [9,10,11,12]. Despite the benefits, most Korean knee OA adults do not meet the frequency recommendations of 150 min aerobic exercise weekly or two times/week strength exercise [4]. When knee OA is combined with comorbidity, the level of physical activity is reduced further [7]; only 9.6 percent of knee OA adults with comorbidity engaged in recommended physical activity [13]. Less physical activity brings an increase of weight, body mass index, or waist circumstance that is a factor exacerbating comorbid chronic disease including diabetes, hypertension, and dyslipidemia [4]. Living with long-term chronic disease can add complexity to performing physical activity with knee OA [7]. 

For healthcare professionals’ effective and practical management of knee OA, it is important to understand barriers to and/or facilitators of physical activity among Korean female adults with knee OA and comorbidity. A sensitive understanding of lifestyles with low levels of physical activity is important for intervention on behalf of Korean female adults with knee OA and comorbidity.

Discovering knee OA patients’ perceptions of their experiences of physical activity through a qualitative rather than quantitative method provides data for clinicians to effectively intervene with management programs. Qualitative methods allow researchers to investigate individuals’ lives, understand the lived experiences, and describe the phenomena [14]. In previous qualitative studies on knee OA patients’ experiences of physical activity, the focus was on their perceptions of pain [15] or surgery [16]. These studies are limited in their ability to obtain information about the influences of comorbidity on physical activity in knee OA patients because comorbidities present new barriers to physical activity after arthroplasty [17]. Previous studies on chronic pain with comorbidity found that comorbid patients believed that healthcare professionals had less understanding and knowledge about comorbidity than about pain and, thus, they did not provide appropriate interventions for them [18,19]. Therefore, this study aimed to explore comorbid knee OA adults’ experiences among Korean women with barriers to and facilitators of physical activity performance.

### Research Question

What are the barriers to and facilitators of physical activity among Korean female adults with knee OA and comorbidity?

## 2. Materials and Methods

### 2.1. Design

This study employed a qualitative content analysis for descriptive exploration. The purpose of this study was to identify and describe common issues and broad characteristics of the phenomenon of interest. This method allows for broad interpretations across multiple relevant fields in nursing and healthcare [20,21,22].

### 2.2. Study Participants

This study was conducted at an orthopedic outpatient clinic of a national university hospital in Cheongju, Chungbuk province, South Korea. The participants were recruited from the orthopedic outpatient population dwelling in their private homes in suburban or rural communities. Purposive sampling was used to include the patients who were the most articulate about their experiences related to knee OA and comorbidity [23]. The inclusion criteria for participation were: (1) Korean female adult diagnosed with knee OA; (2) also diagnosed with one or more comorbid conditions that have a higher frequency in Korean knee OA patients including hypertension, diabetes, or dyslipidemia [3]; and (3) only under medication treatment for knee OA. Individuals who had experienced arthroplasty were excluded because of their differences in physical activity levels and pain intensity [18], and any other surgery was excluded. When data saturation was reached [24], the sampling was ended with 10 participants (Table 1).

Ten Participants with a mean age of 70.7 years (range 56–88) participated in this study. Knee OA had been diagnosed an average of 9.4 years before the study (range: 1–30 years). The types of comorbidity were hypertension, diabetes, and dyslipidemia. The mean knee OA pain intensity score was 8.0 (range: 5–10).

### 2.3. Data Collection

The data were collected between June and August of 2017. Participants were recruited by an orthopedist who was a member of the research team. Demographic characteristics were collected by a research assistant and included age, disease duration of knee OA, types of comorbidity, treatment methods of knee OA, types of physical activity in a recent month, and pain scores without OA medication on knee OA in a recent month. Face-to-face in-depth interviews were conducted during the participants’ scheduled outpatient appointments by the corresponding author, who had previous experience conducting qualitative research and interviews. All interviews were recorded with a digital voice recorder in private and quiet clinical outpatient rooms.

Based on a literature review relevant to the purpose of this study [15,25], semi-structured interview questions were developed to explore the participants’ experiences with barriers to and facilitators of physical activity. The initial interview questions were: “What is your physical activity experience with knee osteoarthritic pain and comorbidity?” “What were the barriers and facilitators for performing the physical activity with pain and comorbidity?” and “What are the results of your physical activity performance?” Based on the responses to these questions, follow-up exploratory questions were asked to obtain more details [26], such as “Can you provide more explanation or examples?” Memos were taken during the interviews to record the participants’ nonverbal communications. After each interview, the interviewer recorded the feeling and ideas expressed in the interview using field notes [27]. The interviewer had an empathic and open mind to avoid researcher bias [28]. The participants were encouraged to freely speak about their experiences without interference from the interviewer. The interviews lasted between 20 and 70 min, and each participant had one or two interviews. This process ended when data saturation was achieved.

### 2.4. Analysis

The data were analyzed using a conventional content analysis method, an inductive analytic process [20,21,22] that we used to describe and learn about the participants’ underlying attitudes toward physical activity, their knee OA pain, and comorbidity. First, the recorded interviews were transcribed verbatim and crosschecked several times by two members of the research team. The analysis encompassed all of the transcribed contents which were considered contextual units for revealing meaning, comprising words, sentences, and paragraphs. To interpret hidden meanings in the context, the two researchers also considered nonverbal communication, such as sighs, silence, posture, tone of voice, and so on. The two researchers read the transcribed texts many times to become immersed in the data and understand the general ideas regarding the participants’ physical activity experiences.

The second analytical phase proceeded through the following steps: coding, creating categories, and abstraction. During coding, the two researchers separately underlined and wrote notes on the page margins next to the meaningful statements about the participants’ physical activity. Then, they independently analyzed the coding and freely generated subcategories for the meaningful statements. While they were creating the subcategories, the two researchers conferred together to reach consensus. In the abstraction step, similar subcategories were merged into categories, and various perspectives were identified and coded through the consensus obtained during those discussions. The two researchers organized the subcategories into categories, and the third author, who is an orthopedic knee specialist highly experienced in treating OA pain and physical activity, provided an expert opinion on the grouping and abstraction of subcategories and categories. All research team members were in complete agreement about the subcategories and categories.

### 2.5. Ethical Considerations

The Institutional Review Board of the corresponding author’s university approved this study (CBNU-201701-BMETC-0132). Before each interview, the patients were given written information and detailed explanations of the study’s purpose and procedures. It was confirmed that they fully understood their rights to confidentiality and anonymity and that they were free to withdraw from the study at any time. All of the patients who confirmed their informed consent were included as participants in the study.

### 2.6. Rigor

The study’s rigor was confirmed using the trustworthiness criteria of Sandelowski [29]. To increase credibility, the data were separately coded and interpreted by two members of the research team. A third member of the research team (with a different disciplinary background) was tasked to limit systematic bias, and the data results were accepted for analysis by the entire team. Two of the study’s participants reviewed the data for accuracy and representativeness by comparing the data to their personal experiences. The interviewer observed the nonverbal communications during the interviews and recorded the observations in the field notes taken during the interviews. Fittingness was satisfied by in-depth assessment until data saturation was achieved. Thick descriptions of the results with representative quotations were chosen to fit the context. Auditability was established through consistent data collection and continuous analytical decision-making. Blanketing ensured the data’s confirmability.

## 3. Results

Four categories of barriers and three of facilitators are derived from the data (Table 2).

### 3.1. Barriers to Physical Activity

#### 3.1.1. Physical Hardships

Intolerable twinges of pain. All participants generally stressed the basic difficulties of physical activity due to intolerable or sleep-depriving levels of knee OA pain. Some types of pain that were difficult for them to clearly articulate were as high as 10 units of intensity and were at the point where participants felt compelled to take analgesic medications for sleep. This amount of pain encouraged many of them to contemplate a surgical solution.
“It’s unbearably painful, itchy, and numbing. Honestly, it’s hard to describe what it actually feels like, whether it’s just itchy, aching painfully and all. It’s almost as if bugs start to creep over your body, and taking a pill (of the prescribed medicine) in the middle of sleep does relieve it. Taking it when eating dinner isn’t enough and it’s necessary to take additional ones while sleeping. With the aching going on every day, it’s really becoming a problem whether to apply for surgery or not.” (Participant 4).

The sense of stiffness in the knees related to the knee OA pain had a negative influence on the participants’ work activities because they were unable to travel long distances for long periods. Further, they reported that they had been actively exercising, traveling, or spending time on hobbies as previous treatments for chronic diseases. However, the participants explained that they no longer were able to perform these activities because of the knee OA, and they inevitably would have experienced a vicious cycle in which the joint pain would continuously increase as the joints deteriorated.
“I used to walk around an hour or so because they said that my legs would stop aching if I exercised. An hour of walking, some biking, and a lot of other activities, but now … it’s just painful. As I try more and more, more pain comes in, and I just simply cannot go on any longer. I know I should exercise, but the sickness ruins everything and I just can’t. That’s probably what makes me sicker, unfortunately.” (Participant 6)

Suffering from excessive physical activity. Many of the participants reported that walking for many hours at a fast pace or exceeding their maximum abilities caused them to suffer at the end of the day from the accumulated pain. Participants told us that their joints signaled pain throughout their bodies when they tried to move beyond their range of motion. In particular, this type of pain tended to be minor or endurable during physical exercise, but it was severely painful at night because of the surge of amassed pain, which interfered with sleep.
“After walking, there was a lot of pain in my muscles that makes me less active these days. I practically do nothing in my home and if I have worked to do, that night’s pain is far worse, especially in the legs. If I work beyond what I can do, all of a sudden my body’s aching.” (Participant 4)

Fatigue and lack of vigor. Three participants stated that they easily felt fatigued, lethargic, and exhausted, which prominently functioned in limiting their abilities to be active. In particular, the older participants suggested that this was the most important reason, along with the excruciating pain, that they were not physically active.
“It becomes something beyond my strength. I just can’t walk. I just don’t have the strength to do so, and if I try to walk to the point where my body’s all sweaty, it’s too tiring. It’s become something I can’t do now, with no energy in my body. My first and foremost reason is that I don’t have any energy, it’s too difficult to do when my body has no energy.” (Participant 2)

Aging. Some of the participants were well aware that physical activity was vital to their health and well-being and that the constraints of aging were gradually decreasing their manageable walking distance. They blamed aging for obstructing physical activity and said they could not help decreasing physical activity because of their age.
“I need to become more active, but I can’t walk much, rather only a bit down the streets. I’m too old to walk long distances compared to about four or five years ago when my body started to fail to do much exercise. It all boils down to aging, of course.” (Participant 6)

#### 3.1.2. Lack of Motivation

Making excuses. Three participants stated that they knew they should engage in some physical activity, but that it was troublesome. They used bad weather like rain or work responsibilities as a pretext to avoid physical activity. They reported instances in which they had urged themselves to be physically active, but they remained supine all day.
“I used the rain as an excuse. I kept moving around in the interior space, but the urge to exercise never really left my thought, just coming and going inside. I do think though, that I actually “need to move.” But, that’s just my mind telling me when my body’s not responding, and I naturally just don’t exercise.” (Participant 8)

Worrying about potential injury. One participant explained that she tended to be more careful about her body parts with OA, and the crookedness of her standing postures seemed to have made it more difficult to maintain her balance in moving vehicles, such as buses. This instability might easily cause accidents involving falls that most likely would injure the body. She stated that she was discouraged from basic physical activity for this reason.
“Walking is hard, with all the pain and the crooked posture, frequently tripping over and bruising my knees, having my skin scraped off. I can’t even ride the bus properly. One second I am standing there, and the next my body starts to fall. Sometimes, when the seats are all taken, I have to stand until I reach my destination. That’s why, on rainy days or when the bus is crowded with people, I don’t get on it. More likely, I can’t get on it, for my body’s sake.” (Participant 10)

#### 3.1.3. Environmental Restrictions

Time constraints. Two participants mentioned that they were advised to exercise or perform physical activities to treat their comorbidities, but that they often faced limitations regarding time. These reasons ranged from work demands to caregiving to their grandchildren.
“I’ve heard the need to exercise for my diabetes. But, I’m a busy woman, with four grandchildren to take care of in the early morning. Of course, I don’t have enough time to exercise… going in and out of my house with my children makes time even less available.” (Participant 3)

Limited resources. Two participants explained that they were unable to access appropriate facilities in rural areas that might aid their ability to exercise. As a result, they could not do anything other than walk around or climb stairs because they were readily available. They said they were unsure if these physical activities were appropriate for their physical conditions.
“It’s hard to find a decent gym near my house. I try to think about exercising every now and then, I walk up the stairs when I have time, but… (it’s not easy.)” (Participant 3)

#### 3.1.4. Lack of Knowledge

Ignorance about their comorbidity and about physical activity. Half of the participants knew very little about the influence of the relationship between knee OA and their comorbidities on their bodies and/or had no understanding of any of the associated characteristics and symptoms. For example, a participant with comorbid diabetes often wanted to stop exercising because of the feeling of fatigue caused by the symptoms of diabetes. In addition, the participant who had experienced symptoms of hypoglycemia preferred decreasing her overall levels of physical activity rather than finding ways to manage a healthier lifestyle, which would have been an effective approach.
“I was sweating like rain while I was just going to the senior citizen center and my clothes got wet in the old days. So I don’t know if my sugar level dropped or what I did at that time. After that day, I was scared because I was so weak. That’s why my physical activity has decreased … I don’t know what to watch out for, and I don’t know what to do.” (Participant 6)

Meanwhile, participants with comorbid hypertension and knee OA had not known that physical activity is required to treat chronic diseases and, instead, thought they should just give up.
“I’ve taken my hypertension pills for around 30 years already, but I still don’t consider whether to change my daily routine because of my illness. In fact, it doesn’t even bother me. If somehow I fall down and lose consciousness, I’ll simply fall down.” (Participant 9)

### 3.2. Facilitators of Physical Activity

#### 3.2.1. Pain Management

Taking prescribed medications. The majority of the participants were on medications prescribed by medical doctors to relieve their knee OA pain. Participants with no previous experience with these medications needed to precisely follow the doctors’ instructions to confirm their effectiveness. They reported that the medications had decreased the knee joint pain. In particular, the use of additional medication when the knee joint pain was severe before physical activity made it much easier to bear.
“When I take the drugs, it doesn’t hurt even after an hour of walking. I think I’m totally fine with only taking the pills, instead of receiving an injection and, because I felt much better after taking the medication, now I eat a pill once every morning or so; but, then, sometimes the pain still hurts much. Even when the pain’s not too hurtful, it makes me take another pill when the pain starts crawling back. After that, and with some exercise, that’s when my body really starts to feel better.” (Participant 5)

Applying adjunctive methods. Many of the participants used adjunctive methods, such as medicated patches, physical therapies, protective gear, and canes to relieve their knee OA pain. These methods subsequently improved their abilities to engage in physical activities. Some of the participants received manual therapy, which temporarily decreased the knee joint pain and improved their performance in various physical activities.
“I’ve tried hard with all the physical therapy, drugs, even going out to receive acupuncture, trying to put on some medicated patches well, the patches did make it less painful. It was hurting less indeed, and since I’m not a person who is going to be stylish and all, it’s important to take a cane with me. Walking with this stick really makes it less tiring. I tried it, and it worked. Once, I was surrounded by younger people climbing up a mountain, and they gave a stick to me. They said that I should walk with it, of course, seeing how I struggled to make my way along, that it would be less painful if I walked with it, and it did save some energy for me. Not only that, but my knee protection, it kind of supports the weight put on my legs, though not completely erasing the pain, it does do some good.” (Participant 9)

#### 3.2.2. Self-Control in Physical Activity

Appropriately intense activity. Half of the participants sought out appropriate speeds, distances, and times to perform their walking activities while enduring their knee OA pain. Most of the participants moved or walked at a slow or easy pace for 15 to 30 min. In the time between walks, the participants rested in response to their knee joint pain in order to manage to continue walking.
“That’s why you should walk at a decent rate little by little, just as much as what your individual strength is capable of. I walk a little bit slower, it’s all sweaty even if I walk slowly. Sometimes it’s necessary to rest a bit, just walk slow as you go, never to the point where it’s painful, though. It’s important that I do it at an adequate level. Exercising is all like that. Exercise can be good for you, but bad for you at the same time, so you should keep it at a manageable level, not too much.” (Participant 2)

Regular exercise. Some participants explained that continuously exercising was beneficial to sustaining muscular strength and flexibility while decreasing the knee joint pain. They stressed that frequent exercise to which one was committed or exercising for quite long periods enabled them to continue their activities.
“Riding a bicycle is painful at the beginning. But, if you keep doing it, you get flexible, and that’s why exercise is nice; if you don’t move, your muscle strength becomes weaker as you walk, so it’s still important to keep moving for flexibility, even though it aches a lot. I feel no strength in my legs at all when I first get up, but it becomes fine after a decent amount of exercise.” (Participant 2)

Lifestyle changes. To treat their comorbidity, four participants maintained healthy lifestyles. Most of them were hypertensive and/or diabetic along with the knee OA, as well as diagnosed with hyperlipidemia and abdominal obesity. However, all who were involved in the lifestyle management program had diabetes. Participants with diabetes told us that the habits of regular physical activity and dietary control were necessary to lessen the pressure on the joints caused by overweight.
“I had to exercise for blood sugar control because of diabetes. So I tried to be physically active (for weight loss) as much as possible. And I tried not to eat a lot while focusing on healthy diets.” (Participant 8)

Decisiveness. Some of this study’s participants generally had personalities that combated idleness or hesitation when performing activities. Despite the aching pain associated with knee OA, these participants displayed a set of decisive behaviors, such as taking the lead when walking, and neither delaying nor hesitating. Overall, they acted with determination when given tasks to accomplish.
“I can’t lay down. I still think I’ve got to do my job, even when I’m crawling on the floor all weak and painful. It’s not like you have to depend on somebody else when you’re sick. I should at least do what I can do myself, independently.” (Participant 10)

#### 3.2.3. Understanding the Importance of Physical Activity

Willingness to engage in physical activities. Half of the participants tended to be internally motivated to engage in voluntary physical activities, such as walking and various exercises. Because they understood that the joint pain increased whether or not the joint moved, the idea that moving would be better for them than sitting tended to induce their voluntary engagement in physical activities. The participants stressed that idleness was annoying and that lying down might increase the pain because of the presence of distracting thoughts.
“It hurts a bit, but I still do a lot of activities, like going here and there… Even if it hurts, I go where I can go. I think it’s important to deliberately exercise a lot. Moving more and more is good … at least it’s better than sitting down and doing nothing.” (Participant 3)

## 4. Discussion

This study was conducted to identify the barriers to and facilitators of physical activity of Korean female adults with knee OA and comorbidity. Overall, participants expressed four categories of barriers and three categories of facilitators.

### 4.1. Barriers

Four barriers were categorized using data from interviews with ten knee OA patients. Hardships were caused by intolerable twinges of pain, suffering from excessive physical activity, fatigue and lack of vigor, and age. The participants perceived these hardships as knee joint pain-related obstacles to physical activity. Knee joint pain with tenderness and stiffness is one of the most frequent complaints of knee OA patients, which leads to serious limitations on their physical activity [11]. Further, knee OA patients might experience increased physical limitations related to aging and comorbidity, such as diabetes [9]. In addition, the knee OA participants in our study reported exhaustion, lack of vigor, and pain after physical activity, all to the extent that they were unlikely to seek exercise opportunities.

Previous studies have suggested that knee OA symptoms, including pain, might be key motivational drivers of physical exercise to reduce this pain, as well as barriers to physical activity [9,25]. Therefore, education and guidelines for healthcare professionals should specifically address physical activity levels and/or intensities [11,25] to improve knee OA patients’ participation and joint pain reduction [9]. As a result, in response to pain or fatigue, people with OA are likely to modify their activities, not by decreasing frequency, but by adjusting intensity [25]. Age was another perceived physical hardship barrier to physical activity, particularly among the older participants, which supports the results of previous studies that found older adults tended not to actively manage their pain because they perceived age-related changes as unavoidable [30]. Interestingly, in contrast with previous studies of barriers to physical activity in OA patients [15,31,32], participants in this study recognized aging as a hindrance to physical activity. This perception may reduce motivation towards physical activity in knee OA patients, which in turn may lead to decreased physical activity. Accordingly, subjective aging perceived by individuals may be a major obstacle regardless of actual age. Therefore, healthcare professionals should encourage older adults with knee OA to make lifestyle changes to avoid risks related to their physical limitations or disabilities [33].

Second, barriers included lack of motivation towards physical activity. The participants made excuses to avoid exercise because of bad weather or work responsibilities. Lack of motivation to stay active, based on laziness or low priority, might lead to pain management failure [9,33], whereas behavioral changes, which would be likely to consume considerable amounts of time and energy, would be relatively difficult [11]. Additionally, patients worried about being injured, which related to postural crookedness, which, in turn, weakened their motivation to engage in physical activity and increased their pain [33]. Thus, healthcare professionals should include motivational factors, such as enjoyment, to develop user-friendly physical activity guidelines for knee OA patients with comorbidity who have low levels of motivation [9].

Third, participants perceived environmental restrictions as barriers with respect to time constraints and limited resources. As a perceived barrier, reported lack of time is consistent with the results of previous studies [33,34] that found people with chronic pain were reluctant to actively exercise because they thought they were too busy. Limited access to facilities is a barrier found by other studies of people with OA [25]. These perceived limitations might lead people to avoid physical activity despite their belief that they need it.

Last, lack of knowledge about their comorbid conditions and about physical activity was a barrier. It was previously suggested that increasing OA severity with increasing comorbidity (diabetes, hypertension, and/or dyslipidemia) positively relates to pain and physical disability [8]. OA and a chronic condition such as metabolic syndrome might be related to chronic inflammation [1,2,8]. Consequently, inaccurate, insufficient, or no knowledge about the relationships of OA to other chronic diseases (including metabolic syndrome) might negatively influence people with OA who might benefit from lifestyle management changes, such as adding regular and/or appropriate physical activity. When insufficient information hinders the prevention of serious complications, a significant barrier might emerge that prevents lifestyle changes that might reduce OA pain and comorbidity. Accordingly, in light of the links between OA and comorbidity, healthcare professionals should provide OA patients with information about their comorbidity (or its potential) and precise, rather than general or vague, information about it [2]. These professionals should be taught the risk factors for developing or increasing OA symptoms based on the patients’ personal experiences [9].

### 4.2. Facilitators

Three facilitators of physical activity were identified among our participants. The most important facilitator was effective pain management, which seemed to be a key enabler of physical activity through medications and adjunctive methods. Although knee joint pain was an obvious main obstacle to physical activity, adjustable pain was a motivator of regular exercise and physical activity among the participants. However, OA sufferers tend to incorrectly use pain medications, so they often experience pain and expend energy on incorrect measures to reduce pain with the goal of reducing their activity and mobility limitations [35]. Therefore, healthcare professionals should educate comorbid knee OA patients on the correct uses of pain medications and supplements [11].

In addition, the participants tended to appropriately adjust their activities to meet their abilities and levels of physical functioning. The participants perceived their activity levels as very low, but no measures of the barriers to and facilitators of physical activity and exercise exist on which knee OA patients can be rated [9]. Thus, healthcare professionals should provide individualized precise guidance to patients based on their particular situations and the factors that influence their physical activities [9]. With tailored guidance, OA patients might adapt their activities by modifying the intensity or by modifying other facilitators identified by this study. This implication relates to Wilcox et al. [25], who found that exercisers tended to adjust the type or intensity of their exercise, whereas non-exercisers tended to avoid exercise.

Considering individual preferences and gradual approaches [8], regular and moderate low-impact physical activities or exercise are important because they maintain healthy joints, strengthen muscles, and prevent excessive joint use and mechanical stress, which effectively reduces pain [2,9,10]. In addition to regular exercise, we found that lifestyle changes were crucial to increasing the amount of the participants’ physical activity. Appropriate lifestyle changes in daily nutrition, habits, work, and leisure are important to the self-management and non-pharmacological treatment of knee OA patients with comorbidities [2,9].

Understanding the importance of physical activity was also identified as an enabler. The participants’ willingness to engage in it was an essential factor for increasing their physical activity. Internal (voluntary) motivation positively influenced active participation in pain reduction and increased physical exercise [33]. Accordingly, healthcare professionals should develop specific ways to motivate comorbid knee OA patients to improve their physical activities and change their lifestyles.

One of the most important findings of this study was that the participants did not indicate any social or environmental facilitators of physical exercise. This is a different result from previous studies on facilitators of physical activity in OA patients [15,31,32]. Thus, healthcare professionals should include social support (such as friends, family, and healthcare workers’ encouragement) and environmental factors (easy access and available facilities/places) when they try to enhance their patients’ physical activities and achieve medical and institutional compliance.

### 4.3. Limitations

This study had some limitations. The participants were all Korean female adults with comorbid knee OA and, therefore, the findings should be interpreted for other cultures and contexts with caution because of the low generalizability of qualitative studies [36]. Future research should consider cultural differences in the factors that influence barriers to and facilitators of physical activity among knee OA patients with comorbidity. In addition, the data were derived from individual interviews that provided patients’ perspectives and further investigation is needed to include healthcare professionals’ viewpoints, which are likely to differ from those of their patients regarding some aspects of physical activity based on patients’ physical functionality. Further research may be considered to carry out a mixed study using both quantitative and qualitative methodology.

## 5. Conclusions

This study contributes to our understanding of comorbid knee OA female adult patients’ perceptions of barriers to and facilitators of physical activity. Knee OA adults with comorbidity are most likely to be older adults with two or more comorbidities, and tailored lifestyle management that links knee OA treatment with other treatments might be more effective when age and type of comorbidity are considered. Therefore, these findings offer healthcare professionals who care for comorbid knee OA patients some important insights into their patients’ needs for appropriate physical activity.

## Figures and Tables

**Table 1 healthcare-08-00226-t001:** Demographic characteristics of participants (*n* = 10).

Participant Number	Age (years)	Duration of Knee OA (years)	Type of Comorbidities	Pain Score of Knee OA (NRS)	Type of Physical Activity
1	64	5	HTN, Dyslipidemia	8	Walking
2	88	20	HTN, Dyslipidemia	5	Walking
3	63	4	HTN, Diabetes	7	Walking, Light climbing
4	77	10	HTN	9	Walking
5	69	10	HTN, Fatty liver	9	Waking, Mild strength exercise
6	75	30	HTN, Diabetes, Dyslipidemia	9	Walking
7	56	1	HTN, Diabetes	9	Walking
8	75	10	HTN, Diabetes, Dyslipidemia	9	Walking
9	74	1	HTN, Diabetes	5	Walking
10	66	3	HTN	10	Walking

OA: osteoarthritis; HTN: hypertension; NRS: Numeric rating scale.

**Table 2 healthcare-08-00226-t002:** Barriers to and facilitators of physical activity among Korean osteoarthritis adults with comorbidity.

Categories	Subcategories	Frequencies *	Participant Number
**Barriers**			
Physical hardships	Intolerable twinges of pain	10	1, 2, 3, 4, 5, 6, 7, 8, 9, 10
	Suffering from excessive physical activity	6	2, 4, 5, 7, 8, 9
	Fatigue and lack of vigor	3	2, 7, 9
	Aging	2	2, 6
Lack of motivation	Making excuses	3	2, 5, 8
	Worrying about potential injury	1	10
Environmental restrictions	Time constraints	2	2, 3
	Limited resources	2	3, 7
Lack of knowledge	Ignorance about their comorbidity and about physical activity	5	1, 3, 6, 9, 10
**Facilitators**			
Pain management	Taking prescribed medications	8	2, 3, 5, 6, 7, 8, 9, 10
	Applying adjunctive methods	6	1, 2, 3, 6, 9, 10
Self-control in physical activity	Appropriately intense activity	5	1, 2, 4, 7, 9
	Regular exercise	3	2, 5, 7
	Lifestyle changes	4	2, 5, 7, 9
	Decisiveness	3	2, 9, 10
Understanding the importance of physical activity	Willingness to engage in physical activity	5	2, 3, 5, 9, 10

* Frequencies mean total numbers of participants in each subcategory.

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
