# Peer review of "Barriers to and Facilitators of Physical Activity among Korean Female Adults with Knee Osteoarthritis and Comorbidity: A Qualitative Study"

_healthcare, 2020, doi:10.3390/healthcare8030226_

Round 1

Reviewer 1 Report

The article is of great interest to the scientific community, but I consider that for its publication, some aspects should be modified, included and polished.

Authors should consider removing the male subject from the sample, as there are 10 women and only one man. Likewise, they should consider the maximum age of analysis, due to the age difference in the sample, they will not have the same barriers and motivations towards the practice of physical activity, a 42-year-old person as an 88-year-old person, are inclusion criteria that they should have been considered in the study design.

In addition, other inclusion criteria must be taken into account, such as practicing physical activity (practice time per week and type of practice). I would like to be indicated. I also ask if they belong to nursing homes or live in their private home, since that determines the answer, as well as whether it is a rural or urban area, since the studies show different results based on these intrinsic characteristics in the sample.

The method should indicate the content methodology applied to analyze the data.

They should add in the results section, the frequencies of the barriers and the facilitators of sports practice.

I have missed the study hypotheses and organizing the discussion based on them.

I would add as a study prospect to carry out a mixed study that used both quantitative and qualitative methodology.

Reviewer 2 Report

This manuscript addressed barriers and facilitators of physical activity in people with OA and additional comorbidities. Below are my comments.

Introduction needs to be re-written overall. I believe the following questions need to be addressed. In the first paragraph, please provide some more backgrounds such as some statistics of OA in Korean population (how many Korean people have OA, what is the economic and societal burdens to Korean healthcare system, how many among them have comorbid conditions? Why is it important for OA patients to be engaged in physical activity? Why are those comorbid conditions likely to make further barriers to physical activity in OA patients?

For Materials and Methods, why were these conditions (hypertension, diabetes mellitus and dyslipidemia) considered only? Is there a specific reason? What if a patient has one of those but also has some other conditions? Were those patients excluded? Treatment by medication means for OA, hypertension, dyslipidemia, diabetes only? What if a patient had a surgery for other conditions? How were pain scores measured? VAS? What pain is the pain score referring to? Knee or hip pain? Overall pain? OA in where? Knee or Hip? What are the grades? No other demographic or clinical measures? Quality of life, functional scores? More details should be provided for participants. 

For Results, how were the categories set? Did the authors make a decision? Provide some descriptive results. For example, how many patients were fit into a certain category or subcategory? For the quotes from participants, please consider to use quotation marks.

For Discussion, so what are the identified barriers and facilitator of physical activity for OA and comorbid conditions compare to those with OA only? Also, please move the limitation to Discussion.

Round 2

Reviewer 2 Report

Thank the authors for revising the manuscript. The manuscript looks good now. My only suggestion will be final check for minor grammatical errors.